# Open Government Data and the Urban–Rural Income Divide in China: An Exploration of Data Inequalities and Their Consequences

Lu Tan *[ID] and Jingsong Pei

School of Economics and Management, Beijing Jiaotong University, Beijing 100044, China; jspei@bjtu.edu.cn
* Correspondence: 19113011@bjtu.edu.cn

**Abstract:** Amidst excitement for the data revolution's potential benefits, concerns mount over its negative impact as unequal data distribution, access, and use widen disparities between individuals and groups, highlighting the urgent need for advanced theoretical and empirical frameworks. This study investigated the impact of open government data (OGD) on the urban–rural income divide in China. Our theoretical analysis shows that the nonrival nature of data initially widens the urban–rural income divide before eventually mitigating it, resulting in an inverted U-shaped relationship. Using a multiperiod difference-in-differences specification, we found that OGD widened the urban–rural income divide between 2010 and 2019. Furthermore, cities with initially wider urban–rural income divides experienced a greater impact from OGD in expanding this divide. These findings provide valuable insights in the role of open data in addressing income inequality, and contribute to our understanding of data inequalities in the context of the data revolution.

**Keywords:** digital economy; open government data; rural China; data divide; income divide; data for development

## 1. Introduction

Data are growing exponentially in volume, velocity, and variety, creating unprecedented opportunities to inform and transform society. Governments, businesses, researchers, and citizen organizations are all exploring, innovating, and adapting to the new world of data [1]. However, data also raise critical questions about access, value, and impact. Who benefits from data and who is left behind? How does data production and use shape the surrounding world? To address these questions, interdisciplinary studies have emerged to examine the social and economic implications of data. They believe that the practices involved in producing, accumulating, and analyzing data have significant consequences on inequality of opportunity and harm [2–4]. While a longer history of research on information and digital inequalities is useful, the data revolution is also producing new forms of inequality that are not easily captured by traditional concepts of digital inequalities and the digital divide. A deeper engagement with critical standpoints in research and practice could help to address these.

In this article, we closely engage with a manifestation of this inequality that can be observed in China, where a salient characteristic is the dichotomy between urban and rural areas. This gives rise to the emergence of an urban–rural divide, wherein the progress of the data revolution in rural sectors lags significantly behind that of urban sectors. We argue that there are positive correlations between open access to data and rural–urban income inequality based on open government data (OGD) in China. We propose an open government data and income inequality model and examine the economic effects of data opening from the perspective of the urban–rural income gap. Our analysis expands the connotation of the urban–rural digital divide to a new generation of data-driven digital

divide and provides policy recommendations for accelerating the balanced development of digital villages and cities to narrow the digital divide.

In the context of the data revolution and data for development (D4D), this paper addresses the issue of data inequality by focusing on a more specific topic: the consequences of open government data on the urban–rural income divide in China. This paper makes several marginal contributions to the literature in the fields of economics of data and critical data studies.

Firstly, while most relevant studies attempt to fit data inequality into broader ICT gaps and digital divides (such as mobile phones, computers, the Internet, and e-commerce), conventional understandings of digital inequalities are not always sufficient to explain and address some causes, forms, and consequences of emerging inequalities resulting from the data revolution [5,6]. Therefore, this article examines diverse data inequalities, ranging from imbalanced data production and unequal data access and utilization by different social groups to income distribution. We attempt to distill the divide into an economic model of data production and income distribution, whereby data inequalities between urban and rural sectors are shaped accumulatively.

Secondly, the difficulty in measuring data amounts and distinguishing the effect of data from the overall technology effect hinders most current studies from devoting sufficient resources towards deeper engagement with evaluating data inequality. In this article, we attempt to address this shortage by studying a prototypical setting: open government data in China and its consequence on the urban–rural income divide using multiperiod difference-in-differences estimation.

Section 2 provides a comprehensive review of the literature, highlighting the concerning impact of technological progress and the economics of data in a digital age. Building on this, in Section 3, we develop an economic model to examine the effects of open data access on the urban–rural income divide. Our model yields a critical prediction that guides our subsequent empirical analysis. In Section 4, we describe our data sources, variables, and methodology, which employs a difference-in-differences estimation approach. We exploit the exogenous variation in the timing of government data openness across cities and years to assess its causal impact on the urban–rural income divide. Section 5 presents our econometric results and robustness tests. Through this analysis, we contribute to the literature regarding the impact of open data access on the urban–rural income divide, which has important implications for policy and economic development.

## 2. Related Literature

While the exact extent of rural–urban income disparity in China is debated due to discrepancies in data sources and methods, it is generally acknowledged to be significant [7], with China having one of the highest urban–rural income ratios globally [8,9], and numerous studies have investigated this divide. The urban–rural income gap accounts for a substantial portion of China's overall income inequality, ranging from over 50 percent in most studies to approximately one quarter [9]. Xie and Zhou found that this gap contributes to over 10 percent of total national inequality in China but is negligible in the United States [10]. Overall, the majority of studies indicate that urban–rural income disparity is the primary contributor to China's income gap [11,12].

Many studies have attempted to identify the fundamental causes of rising income inequality in China. The most often suggested, tested, and examined causes are as follows. First, early political strategies promoted heavy industries and manufacturing [13]. Consequently, urban expansion was fostered by investments, favorable policies, and financial aid. Agriculture lagged and was utilized to build other industries. Urban capital buildup and subsidies were funded by agricultural surplus. State control of agricultural output and procurement, food price reduction, and Hukou system limits on rural–urban movement harmed rural inhabitants' incomes.

Second, industrial restructuring increases the conflicts between state industry and staple agriculture. Since the reform of the economy in 1978, China's primary, secondary,

and tertiary industries have changed from 28 percent, 48 percent, and 24 percent of GDP in 1978 to 7 percent, 41 percent, and 52 percent in 2018, respectively (China Statistical Yearbook, 2019). The relationship between industrialization and income change has been extensively studied [14].

Third, the influx of rural residents into cities drives urbanization, allowing for the flow of people, skills, capital, goods, and information, which contributes to the urban–rural income gap [15]. Li's survey data suggest that rural migration boosts rural income by enhancing labor productivity of migrant workers and improving allocation of nonmigrant workers [16]. Urbanization may also narrow the urban–rural income gap.

The development of financial sectors may be the fourth reason for income disparity. Studies show an inverted U-shaped relationship between financial development and income distribution, as well as a significant widening of the urban–rural income gap due to financial intermediary evolution [17]. There is also positive and negative causality between financial development and efficiency and the urban–rural income gap [18,19].

Scholars have also looked at the gap through the lens of human capital. Guo found that low human capital accumulation in rural areas leads to stagnant income growth, while urban areas have more sustainable and balanced growth with high human capital accumulation [20].

The effect of China's opening up reforms on urban–rural income inequality is inconclusive. Wei and Yi found that the reforms narrowed the gap in around 100 cities [21], while Hertel and Fan argued that WTO accession and market reforms reduced inequality [22]. In contrast, Jeanneney and Hua found that the exchange rate policy affected inequality [23], and Wei and Zhao proved that international trade increased the gap through employment and remuneration effects [24].

Finally, the impact of technological progress on the income divide has been extensively studied and discussed by scholars, particularly in the era of the "new economy". Acemoglu argues that the technological progress represented by information technology in the third industrial revolution is generally skill-biased, which can explain the wage divide between skilled labor and unskilled labor in recent decades [25]. Regarding the income gap between urban and rural areas, Zou and Liu find that there is a matching gap between the agricultural skills possessed by rural residents and the industrial skills required by the urban industrial sector, leading to lower wages in informal employment [26]. According to Acemoglu and Aghion [25,27,28], the skill bias of technological progress polarizes remuneration growth for high-skilled and low-skilled labor. This is the same factor that exacerbates China's urban–rural income inequality. The Kuznets inverted U hypothesis has been extensively applied to analyze the impact of technology on this inequality [29–31]. The literature suggests that technology can lead to an initial increase in income inequality due to the concentration of technological innovation in urban areas, but, after a certain threshold, further technological progress can lead to a decrease in income inequality due to the diffusion of technology to rural areas. Conversely, Ji et al. argue that technological progress in China widens the income gap rather than narrowing it, contradicting Kuznets' inverted U hypothesis [32]. Li et al. and Luo and Hu find a positive correlation between technological progress and the urban–rural income gap, with knowledge capitalization being a significant contributor [33,34]. However, as Acemoglu notes, different technologies have different effects due to their nature, and the "data" concerned in this paper are different from the technological progress in most of the current literature [25]. Thus, the impact of differentiation, especially on the urban–rural income gap, deserves attention as a critical research problem in economics.

The interdisciplinary studies examining the social and economic implications of data are rapidly growing, and this paper aims to provide a perspective on the inequality that arises from open data. Our focus has similarities with the literature on information economics, the data market, and data access. In particular, the concept of data as an input of economic activity has been studied in the literature on information economics. For example, Refs. [35,36] view data as a form of labor, and highlight that people may not be

adequately compensated for the data they provide. Our analysis of open data access effects centers on a market for data, which is related to the market for ideas in [37]. However, unlike the market for ideas, where only one firm can use an idea at a time, our market for data enables multiple firms to use nonrival data simultaneously. It is closest to the literature of Ali et al [38], which examines the sale of nonrival information in a search and matching decentralized market and demonstrates that nonrivalry can result in inefficiency due to the underutilization of information. Moreover, Ichihashi studied competition among data intermediaries [39], while Akcigit and Liu investigated the social value of sharing information across firms in a growth context [40].

Although our paper emphasizes the effect of data access on labor, most of the existing literature is interested in the effect of data access on industrial organization. Varian provides a general discussion on the economics of data and machine learning, and notes that data are nonrival, suggesting that data access may be more important than data ownership [41]. Farboodi and Veldkamp explore the implications of expanding access to data for financial markets [42], while Farboodi and Veldkamp suggest that access to big data has led to a rise in firm size inequality [43]. Finally, Hughes-Cromwick and Coronado view government data as a public good and study their value to US businesses [44].

### 3. Concept, Background, and Model of Data

#### 3.1. Data Concept

The literature frequently confuses the three terms "data", "information", and "knowledge" due to their overlapping usage. Hence, to establish the conceptual boundaries of "data", it is important to understand the distinctions among the three. In essence, the term "data" refers to raw facts and figures, which in turn convey "information". The assimilation of "information" generates "knowledge" at a collective level within a specific social context [45], which can fuel innovation and contribute to improvements in productivity and services. Moreover, knowledge spillovers have emerged as a significant driver of economic growth, particularly in situations where resources are scarce [46]. In contrast to "information", which refers to meaningful, structured data, "data" are often meaningless and unstructured [47]. Extracting information from data requires specialized tools and techniques collectively referred to as data analytics [48].

#### 3.2. Economic Nature of Data

Scholars often compare "data" to oil, as evidenced by [15,49,50]. Unlike oil, however, data are nonrival, which implies that they are inexhaustible. To generate value, data need to be processed, interpreted, and analyzed by either human beings or automated systems [47], and their usefulness is dependent on application technology and specific scenarios [51].

From an economic and social perspective, data can be classified into various categories, such as public- and private-sector data, open and closed data, personal and nonpersonal data, and user-generated and machine-generated data. Of these, open and closed data are most closely associated with data-driven economic growth and improved living standards. Since data are nonrival, opening data can maximize their application in downstream production, and the spillover effects of knowledge, as suggested by Frischmann, can enhance the economic and social value of data, resulting in increased returns to scale [52].

#### 3.3. Open Government Data (OGD)

The opening of government data in China represents one of the most significant events in data management as it provides private firms access to a vast amount and diverse types of data collected by the government in delivering public services. This open government data (OGD) initiative has the potential to enhance government transparency, enable the public to monitor government performance and hold it accountable for wrongdoings, and create new opportunities for economic growth and social innovation. Private enterprises can utilize OGD to develop new products and services, such as real-time roadway traffic monitoring and business trend analysis, leading to increased income [53,54].

Opening up government data is of particular importance as it can drive knowledge-based economic growth, and this has been supported by recent academic literature [55,56]. Furthermore, research shows that the commercial utilization of OGD can result in a positive impact on the performance and productivity of firms, leading to enhanced competitiveness [57,58].

### 3.4. An OGD and Income Inequality Model

Drawing on the idea of the TKC and the data production model proposed by Jones and Tonetti [59], we develop a model to examine the relationship between open government data (OGD) and income inequality in the rural–urban divide. Given China's unique dual rural–urban structure, we assume that the economic system comprises only two sectors: rural and urban. The rural sector produces $N_r$ varieties of consumer goods, while the urban sector produces $N_u$ varieties. Aggregate output in each sector, $Y_r$ and $Y_u$, is symmetrically expressed in the production function. Therefore, we can construct their production functions as follows:

$$Y_r = \left( \int_0^N Y_{ri}^{\frac{\sigma-1}{\sigma}} \, di \right)^{\frac{\sigma}{\sigma-1}} = N_r^{\frac{\sigma}{\sigma-1}} Y_{ri}, \tag{1}$$

$$Y_u = N_u^{\frac{\tau}{\tau-1}} Y_{ui}. \tag{2}$$

Variety $i$ is produced by combining an idea of quality $A_i$ and labor $L_i$:

$$Y_i = A_i L_i. \tag{3}$$

Since data can be used to train artificial intelligence algorithms, data $D$ can be regarded as used to improve the quality of knowledge $A$:

$$A_i = D_i^\eta, \tag{4}$$

where $\eta$ determines the importance of the data. Jones and Tonetti [59] suggested that $\eta$ might take a value of 0.03 to 0.10, so we require it as a small positive number, much smaller than 1. Substitute (4) into (3), and then the production function of variety $i$ is

$$Y_i = D_i^\eta L_i = D_i^\eta L / N, \tag{5}$$

where $L$ is the total amount of labor in the industry, symmetrically distributed to each variety.

The consumption of a product generates a corresponding set of data. For instance, the operation of a self-driving car produces data for every kilometer traveled, which can be utilized to improve the efficiency of future trips. Furthermore, data on traffic and driving patterns, collected and managed by the government, can be valuable to self-driving companies. It is assumed that there exists a constant proportional relationship between open government data (OGD) for rural and urban sectors, denoted as $x_{rg}$ and $x_{ug}$, respectively, and overall OGD, which is represented by $x_g$. Specifically, rural OGD is $x_{rg} = \theta x_g$ and urban OGD is $x_{ug} = (1-\theta)x_g$. Therefore, the data generated in rural and urban sectors can be expressed as follows:

$$D_{ri} = (x_i Y_{ri})^\alpha \left( \theta x_g N_r Y_{ri} \right)^{1-\alpha} \tag{6}$$

$$D_{ui} = (x_i Y_{ui})^\beta [(1-\theta) x_g N_u Y_{ui}]^{1-\beta}. \tag{7}$$

where $Y_{ri}$ and $Y_{ui}$ denote the amount of data generated by a single firm $i$ in the rural and urban sector, respectively; $x_i$ is the proportion of data available to firm $i$. The quantities $N_r Y_{ri}$ and $N_u Y_{ui}$ are the amounts of data generated by other varieties in the two sectors because variety $i$ is infinitesimal for the firms that are symmetric. Therefore, $\theta x_g N_r Y_{ri}$ and

$(1-\theta)x_g N_u Y_{ui}$ are the amounts of data that can be used simultaneously by any firm in rural and urban sectors, depending on the nonrival nature of data.

Substitute (6) and (7) into the (1) and (2) aggregate output:

$$Y_r = \left[ N_r^{\frac{1+\eta(1-\alpha)(\sigma-1)-\eta\sigma}{\sigma-1}} \left( x_i^\alpha \theta^{1-\alpha} x_g^{1-\alpha} \right)^\eta L_r \right]^{\frac{1}{1-\eta}}. \tag{8}$$

$$Y_u = \left[ N_u^{\frac{1+\eta(1-\beta)(\tau-1)-\eta\tau}{\tau-1}} \left( x_i^\beta (1-\theta)^{1-\beta} x_g^{1-\beta} \right)^\eta L_u \right]^{\frac{1}{1-\eta}}. \tag{9}$$

With a multiplier $\frac{1}{1-\eta}$, the more people who consume the variety, the more data received. This increases productivity and consumes more, which in turn generates more data. Therefore, $\frac{1}{1-\eta}$ is considered the key exponent in the production function. $L_r$ and $L_u$ represent, respectively, labor input of the rural and urban sector. $\alpha$ and $\beta$ are the marginal output elasticity of rural and urban firms, respectively. Assuming that the market is perfectly competitive and the production function has constant returns to scale, the income level of rural and urban residents, $I_r$ and $I_u$, is therefore equal to the marginal income of their labor, which can be calculated as

$$I_r = \frac{\partial Y_r}{\partial L_r} = \frac{1}{1-\eta} N_r^{\frac{1+\eta(1-\alpha)-\eta\sigma}{(\sigma-1)(1-\eta)}} \left( x_i^\alpha \theta^{1-\alpha} x_g^{1-\alpha} \right)^{\frac{\eta}{1-\eta}} L_r^{\frac{\eta}{1-\eta}}, \tag{10}$$

$$I_u = \frac{\partial Y_u}{\partial L_u} = \frac{1}{1-\eta} N_u^{\frac{1+\eta(1-\beta)-\eta\tau}{(\tau-1)(1-\eta)}} \left( x_i^\beta (1-\theta)^{1-\beta} x_g^{1-\beta} \right)^{\frac{\eta}{1-\eta}} L_u^{\frac{\eta}{1-\eta}}. \tag{11}$$

Equations (9) and (10) show that the marginal income of the labor is proportional to the size of the economy in each sector, raised to some power. The exponent, $\frac{\eta}{1-\eta}$, captures the degree of increasing returns to scale in the economy that reflects the increasing returns associated with the nonrivalry among data. It increases in $\eta$, the importance of data to the economy. This confirms the idea that a larger economy is richer because it produces more data, which then feed back and make all firms more productive.

Thus, the rural–urban income gap (GAP) can be measured by

$$GAP = \frac{I_r}{I_u} = \frac{N_r^{\frac{1+\eta(1-\alpha)-\eta\sigma}{(\sigma-1)(1-\eta)}} \left( x_i^\alpha \theta^{1-\alpha} x_g^{1-\alpha} \right)^{\frac{\eta}{1-\eta}} L_r^{\frac{\eta}{1-\eta}}}{N_u^{\frac{1+\eta(1-\beta)-\eta\tau}{(\tau-1)(1-\eta)}} \left( x_i^\beta (1-\theta)^{1-\beta} x_g^{1-\beta} \right)^{\frac{\eta}{1-\eta}} L_u^{\frac{\eta}{1-\eta}}}. \tag{12}$$

Finally, to analyze how the rural–urban income gap has changed in relation to OGD, we take the first derivative of GAP with respect to OGD, which can be expressed as:

$$\frac{\partial GAP}{\partial x_g} = \frac{(\beta-\alpha)\frac{\eta}{1-\eta} N_r^{\frac{1+\eta(1-\alpha)-\eta\sigma}{(\sigma-1)(1-\eta)}} N_u^{\frac{1+\eta(1-\beta)-\eta\tau}{(\tau-1)(1-\eta)}} x_i^{\frac{(\alpha+\beta)\eta}{1-\eta}} \theta^{\frac{(1-\alpha)\eta}{1-\eta}} (1-\theta)^{\frac{(1-\beta)\eta}{1-\eta}} x_g^{\frac{(3-\alpha-\beta)\eta-1}{1-\eta}} L_r^{\frac{\eta}{1-\eta}} L_u^{\frac{\eta}{1-\eta}}}{\left\{ N_u^{\frac{1+\eta(1-\beta)-\eta\tau}{(\tau-1)(1-\eta)}} \left( x_i \beta (1-\theta)^{1-\beta} x_g^{1-\beta} \right)^{\frac{\eta}{1-\eta}} L_u^{\frac{\eta}{1-\eta}} \right\}^2}. \tag{13}$$

where the denominator is greater than zero and the numerator can be positive or negative, depending on the sign of $\beta - \alpha$, since $\frac{\eta}{1-\eta} > 0$ ($0 < \eta < 1$). From Kuznets' view [60], the marginal production elasticity of urban capital is greater than that of rural capital during the early stages of economic development when $\beta > \alpha$ stage. Since production in the digital economy is data-driven, data are endogenous and the stage of economic development depends on them, so that $\frac{\partial GAP}{\partial x_g} > 0$ and OGD worsens the gap; as the economy develops, when $\beta = \alpha$, the rural capital marginal output elasticity equals the marginal output elasticity of urban capital, at which point the gap peaks. Beyond this point, then, $\beta < \alpha$, $\frac{\partial GAP}{\partial x_g} < 0$. Economic development and data drive benefit the rural sector more than the urban sector,

and, finally, OGD decreases. Based on the analysis of the theoretical model above, we can confirm that the data, similarly to other technological advances, have an inverted U-shaped effect on income inequality, meaning that the gap increases and then decreases as OGD grows.

This relationship between OGD and income inequality can be explained through the transition process. As individuals move from the rural sector to the advanced urban sector, OGD would at first improve the income of those moving to the latter but exacerbate income inequality for those who remain in the less developed rural sector. The disparity in the amount of data opened by the government between urban and rural areas in Figure 1 has provided us some clues. However, as the transition process concludes, OGD can narrow the income gap by increasing the data utilization capacity and data amount in the rural sector. This improvement in capacity allows rural communities to benefit from OGD, thereby reducing income inequality.

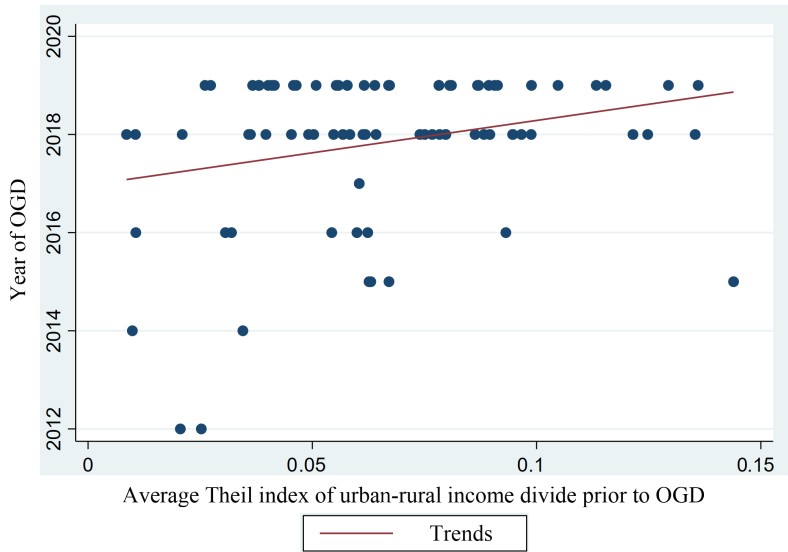

**Figure 1.** Graphical analysis regression of OGD timing and the urban–rural income inequality Theil index.

As China is currently in the process of achieving balanced and sustainable development between its urban and rural sectors and is at the beginning of a data-driven economy, we believe that the relationship between OGD and urban–rural income inequality is experiencing its initial stage. Therefore, we propose the following proposition:

Between 2010 and 2019, there was a positive correlation between the opening of government data and the widening of the urban–rural income divide.

## 4. Data and Methodology

To evaluate the impact of OGD on income disparity between urban and rural sectors, we collected data on the timing of government data availability, as well as information on the distribution of income in both urban and rural regions and other pertinent regional-level characteristics. This section provides an overview of the data that we have gathered, as well as a description of the econometric methods that we employed.

### 4.1. OGD Context in China

In the 1990s, before the liberalization of the planned economy in China, government and individual information were not disclosed to the public. All information was centralized for government use only, and individuals were not active agents in the planned economy. Since liberalization, there has been a transformation in the relationship between individuals and the government, which has increased the power of individuals and made the government more responsive to their needs. As technology has developed and the

government's capability of utilizing it has grown, China has gradually collected and limited the disclosed information of government and individuals to ensure the right to be informed and information equivalence in a market economy [61].

Shanghai was the first city to launch the "Shanghai Municipal Government Data Service Network" in June 2012, and other cities, such as Beijing, Foshan, and Wuhan, have also launched government open data platforms in subsequent years. The Chinese State Council issued the "Action Plan to Promote Big Data Development" in August 2015, stating that "data has become a fundamental strategic resource of the country, and China will gradually promote the opening of government data resources to society". By the first half of 2020, 130 government data opening platforms had been launched, compared to 18 in 2017, 56 in 2018, and 102 in 2019 [62]. Table A1 provides the year in which each city opened its government data.

The Chinese government's 2015 Action Program for Promoting the Development of Big Data mandated the opening of various datasets, primarily in the fields of trade and business, social livelihood, organizations, and medical and health care, as depicted in Figure A1 [54]. In contrast, agricultural and rural areas and credit services have the fewest open datasets. This disparity in datasets between urban and rural areas provides important context for our subsequent research.

The implementation of the Chinese government's open data policy has progressed through three stages, starting with a slow period of development, followed by a period of in-depth development with little noticeable change, and, finally, a period of continuous improvement, leading to an OGD boom beginning in 2014 [63]. By the second half of 2021, many cities had datasets exceeding 100 million, with Dongguan reaching nearly 400 million, and a single average dataset volume exceeding 880,000. These high-capacity datasets are rich in content related to commerce, society, market regulation, law enforcement, and other aspects [64]. Utilizing high-capacity datasets can provide more value and better reflect the level of openness of a dataset.

*4.2. Urban–Rural Income Divide Data*

The income information for rural and urban areas was obtained from various sources, including the China City Statistical Yearbook, China Rural Statistical Yearbook, and local statistical yearbooks of cities and provinces in China. These annual surveys provide extensive data on incomes and other household characteristics across China, making them ideal for capturing substantial variation and analyzing trends over time. While they are nationally representative samplings of the population, they do not track individuals over time. They provide information on the disposable income of urban and rural residents, as well as a wide array of demographic characteristics in the year prior to the surveys.

To measure the urban–rural income divide, we used the Theil index for each city and year. While the ratio of per capita disposable income of urban and rural residents is a commonly used indicator, it cannot fully reflect the impact of changes in the proportion of the urban and rural population. The Gini coefficient is a more rigorous measure, but it requires decomposition among people of different classes and is sensitive to income changes in the upper class. The Theil index, on the other hand, is decomposable to study differences in urban versus rural populations and captures changes at both ends of the income spectrum [65–67]. It represents our study's income divide between urban and rural residents. A larger result indicates a more significant urban–rural income divide.

$$Theil_{i,t} = \int_{i=1}^{2} \left( \frac{Y_{i,t}}{Y_t} \right) \times \ln \left( \frac{\frac{Y_{i,t}}{Y_t}}{\frac{X_{i,t}}{X_t}} \right), \tag{14}$$

where $i = 1$ represents the urban and $i = 2$ represents the rural. $Y_{i,t}$ stands for urban or rural disposable income in the year $t$; $Y_t$ stands for total disposable income in the year $t$; $X_{i,t}$ represents the urban or rural population figures in the year $t$; and $X_t$ represents the total population in the year $t$.

Cities in China are administratively divided into township-level units, which are further subdivided into village-level units for rural areas and district-level units for urban areas. However, these units are not independent of each other, and both rural and urban areas are integral parts of cities. Given this, many studies examining the urban–rural divide in China have used city-level data. To better understand this divide, our study focuses on households in villages and urban neighborhoods under the administration of 326 cities from 2010 to 2019 as most OGD are available at the city level. Our sample includes 2121 city year observations.

In Table A2, we present descriptive statistics on the Theil index of the urban–rural income divide, which is measured at the city year level. In addition to reporting the mean, minimum, and maximum values of the Theil index, we also present the standard deviations of the Theil index across cities, within cities, and within city years. The cross-city standard deviation of Theil is the standard deviation of $(Theil_{ct} - Theil_c)$, where $Theil_c$ is the average value of Theil index in city $c$ over the sample period. The within-city standard deviation of the Theil index is the standard deviation of $(Theil_{ct} - Theil_t)$, where $Theil_t$ is the average value of the Theil index in the year $t$. The within city year standard deviation of the Theil index is the standard deviation of $(Theil_{ct} - Theil_c - Theil_t)$, where $Theil_c$ is the average value of Theil index in city $c$ and $Theil_t$ is the average value of the Theil index in year $t$. These standard deviations help in assessing the economic magnitude of the impact of OGD on the urban–rural income divide.

*4.3. Methodology*

Following the launch of the "Shanghai Municipal Government Data Service Network" in June 2012, Beijing, Foshan Nanhai, Wuhan, and other regions have also launched government open data platforms successively. Since government data were opened in different years in different cities, it provides a natural setting for an event study to use multiperiod difference-in-differences (DID) specification to assess the relation between OGD and urban–rural income divide [68]. As an extension of the standard DID method, multiperiod DID allows for more than two periods of data. It is used to estimate the causal effect of a policy intervention that begins at different times in different cities. In our scenario, the DID method compares the change in outcomes over time between the treatment group (cities that opened government data) and the control group (cities that did not open government data). The regression is set up as follows:

$$Y_ct = \alpha + \beta D_ct + \gamma X_ct + A_c + B_t + \varepsilon_ct, c = 1, \cdots; t = 2010, \cdots, 2019. \tag{15}$$

In Equation (15), $Y_{ct}$ represents Theil index of the income divide between urban and rural residents in city $c$ in year $t$. $A_c$ and $B_t$ are the dummy variables, representing the city and year fixed effects, accurately reflecting the city and year characteristics. $X_{ct}$ is a series of city-level control variables that change over time. These variables were chosen based on the most extensively examined causes of the income divide, as discussed in Section 2. We accounted for changes in a city's economy over time by controlling the following factors: local government influence, changes in industrial structure, level of urbanization, development of financial sectors, opening-up effect, and level of human capital. The variables included in $X_{ct}$ are general public budget expenditure as a proportion of the regional GDP (publicout-gdp), the proportion of employment in secondary and tertiary industry (rindustry2nd, rindustry3rd), urbanization (urban), loans of the national banking system at year end as a proportion of regional GDP (loan-gdp), number of students enrolled at regular institutions of higher education per 100 persons (collegestu-100), amount of foreign capital actually utilized as a proportion of the regional GDP (foreigncapit-gdp). $\varepsilon_{ct}$ is the error term. $D_{ct}$ is the dummy variable in which we are interested. If the government of city $c$ opens its data in the year $t$, the value of $D_{ct}$ equals 1; otherwise, it is 0. The coefficient $\beta$ reflects the impact of OGD on the urban–rural income divide. If $\beta$ is positive and significant, it indicates that OGD promotes the growth of the Theil index; that is, it widens the urban–rural income divide. On the contrary, if $\beta$ is negative and significant, the

divide is narrowed by OGD. The DID model enables us to solve the problem of omitted variables to a large extent. We include year fixed effect dummy variables to control the influence of shocks and trends that change over time on the urban–rural income divide, consisting of the business cycle, trends in income distributions between different classes, and changes in other policies and regulations. Meanwhile, we include city fixed effect dummy variables to control the influence of unobserved time-invariant city characteristics on the divide. (All the calculations were run in Stata).

## 5. OGD and Urban–Rural Income Divide

### 5.1. Preliminary Result

Our empirical analysis is based on the fact that the timing of OGD was unaffected by the income divide. In order to test this premise, we conducted a regression on the OGD timing to the urban–rural income Thiel index (graphical result shown in Figure 1) and its changes (graphical result shown in Figure 2). According to the insignificant results of Figures 1 and 2, before and on the OGD, neither the urban–rural income Thiel index changes nor the Thiel index itself can explain the timing of the OGD. This shows that the timing of OGD does not change with the degree of pre-existing urban–rural income inequality, which is consistent with the basis of the empirical analysis in our study.

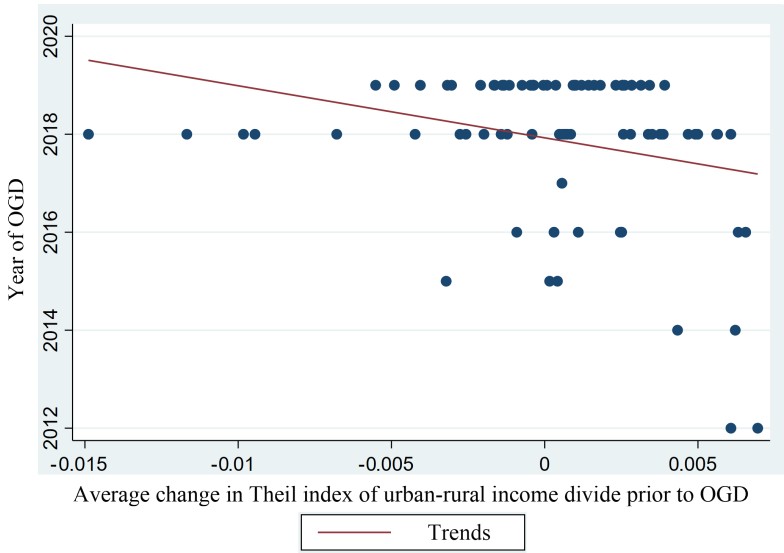

**Figure 2.** Graphical analysis of OGD timing and urban–rural income inequality Thiel index changes. Figure 1 shows a scatter plot of the average Thiel index of urban–rural income divide prior to OGD and the year of OGD. Figure 2 shows a scatter plot of the average change in Thiel index of urban–rural income divide prior to OGD and the year of OGD. According to the regression results, the coefficient is statistically insignificant, and no obvious trend can be found in Figures 1 and 2.

### 5.2. Main Result

The results in Table 1 show that the openness of government data to a certain extent aggravates the urban–rural income divide. We use two regression models to evaluate the impact of government data opening on urban–rural income inequality. In these, the coefficient of government data openness is positive and significant. The second regression results show that, even after controlling for several time-varying city characteristics, OGD is still positively correlated with widening the urban–rural income divide (significant at the 5 percent level). The regression results showed that opening government data in a city would lead to a 0.84 percent increase in the urban–rural income Thiel index without control variables and a 0.58 percent increase with control variables. To measure its economic effect, we compared the coefficient of OGD with the 1.3 percent standard deviation of the Thiel index after accounting for the city and time fixed effects. The standardized coefficient is

64.6 percent without control variables and 44 percent with control variables, suggesting that OGD can explain more than 40 percent of rural–urban income inequality after controlling for changes in the urban–rural income divide caused by city and time effects. The city and year fixed effects explain much more of the total variation in the urban–rural income divide than OGD. To assess potential collinearity effects and ensure the suitability of our model, we employed variance inflation factors (VIFs) for each predictor. The aim was to investigate whether the high R2 values (0.921) were influenced by collinearity. Our findings revealed that all VIF values were below 5, indicating no significant correlation among the predictors. These results provide evidence that open government data (OGD) does indeed influence the income disparity between urban and rural areas.

**Table 1.** Regression results of government data opening on urban–rural income divide.

| | VIF | (1) Theil | (2) Theil |
|---|---|---|---|
| OGD | 1.1193 | 0.0084 *** | 0.0058 ** |
| | | (2.68) | (2.16) |
| publicout-gdp | 1.6865 | | −0.0538 ** |
| | | | (−2.12) |
| rindustry2nd | 1.0291 | | 0.0000 |
| | | | (0.40) |
| rindustry3rd | 1.6854 | | −0.0004 *** |
| | | | (−3.72) |
| urban | 2.3866 | | −0.1824 *** |
| | | | (−2.85) |
| loan-gdp | 1.8877 | | 0.0023 |
| | | | (1.61) |
| collegestu-100 | 1.9854 | | 0.0008 |
| | | | (0.70) |
| foreigncapit-gdp | 1.0054 | | 0.1707 * |
| | | | (1.84) |
| cons | | 0.0839 *** | 0.2039 *** |
| | | (639.47) | (5.60) |
| N | | 2236 | 2102 |
| R$^2$ | | 0.921 | 0.921 |

*t* statistics in parentheses. * $p < 0.1$, ** $p < 0.05$, *** $p < 0.01$.

*5.3. Robustness Tests*

5.3.1. Placebo Test

To assess the possible association of other time-varying city characteristics with the timing of OGD and changes in the urban–rural income divide that may be present if the main analysis is flawed but should not be present otherwise, it is necessary to test the exclusivity of the effect of OGD on the urban–rural income divide. We designed a placebo test by advancing the OGD time for 3 years. The result was insignificant, indicating that the "fake-policy dummy variable" cannot explain the change in urban–rural income divide in the fake scenario, and the change in the explained variable is unlikely to be affected by other policies or random factors, which passes the test.

5.3.2. Bootstrap Test

Since the OGD began from 2012 in China, the number of available samples has been limited. We tried to use bootstrap and SUR standard error estimators to test the results by creating bootstrap samples from the original dataset by sampling with replacement. We randomly extracted the observation results from the data and calculated the impact of government data opening on the urban–rural income divide with the city and time fixed effects. We took 500 such samples and estimated the impact of government data openness on the rural–urban income divide by a factor of 500. The standard deviation of the result estimate was bootstrap. Then, the SUR standard error was estimated and the

nondiagonal elements of the weighted matrix were restricted to be invariant. The result was also significant.

### 5.3.3. Parallel Trends Test

To ensure the results meet the unbiasedness, the trend of parallel hypothesis between the treatment and control groups needs to be met. If the pretreatment trends of the treatment and control groups are different, parallel trends do not hold, which questions whether the income divide changes are not caused by the OGD but by other trends. Therefore, it is necessary to verify whether there is a parallel trend in the income divide of treatment cities and control cities before the OGD.

Just like the classical DID model, multiperiod DID can also be decomposed and analyzed for the dynamic economic effects of policies through the event study method, but the policy time needs to be treated centrally (the time of each period minus the policy implementation time). The regression model is

$$Y_{ct} = \alpha + \sum_{\tau=-M}^{N} \beta_\tau D_{c,t-\tau} + \gamma X_{ct} + A_c + B_t + \varepsilon_{ct}, \tag{16}$$

where $\sum_{\tau=-M}^{N} \beta_\tau D_{c,t-\tau}$ is a dummy variable; $M$ and $N$ are the periods before and after OGD, respectively. If city $c$ opens government data in time $t-\tau$, the dummy variable equals 1; otherwise, 0. For example, when $\tau = 2$, $D_{c,t-\tau}$ represents city $c$ opening government data in time $t - 2$; the regression estimates the effects in the second year after data opening. $\beta_\tau$ represents the current period of opening government data; $\beta_{-M}$ to $\beta_{-1}$ represent the $1 - M$ time period before opening government data; $\beta_1$ to $\beta_N$ represent the $1 - N$ time period after opening government data. If the result is significant and equals 0 in the period of $\beta_{-M}$ to $\beta_{-1}$, it indicates that there is no significant difference between the treatment and control groups in period $1 - M$ before the opening of government data, which asserts that parallel trends hold.

The result in Figure 3 shows that the confidence interval on the left side of 0 includes 0, and the confidence interval on the right side of 0, where the vertical line at each point intersects with the X-axis, departs from 0. It indicates no significant difference (heterogeneous time trend) between the treatment group and the control group before the opening of government data. Our model satisfies the assumption of parallel trends, so the estimated result is unbiased and pure.

According to the main regression and robustness test results above, we can conclude that H1 can be accepted. There is a positive relationship between government data opening and the widening of the urban–rural income divide, and, at the present stage, this relationship is becoming stronger and stronger over time.

### 5.4. Impact of OGD and Initial Conditions

To assess whether the impact of OGD on income inequality varies in predictable ways across cities with different initial conditions, we built a quantile regression model for the initial urban–rural income Theil index with fixed effects. The quantiles selected were 0.2, 0.4, 0.6, and 0.8, respectively. The lower the quantile, the greater the initial income divide, and vice versa.

Table A3 presents the results, showing that OGD has a significant impact on the urban–rural income divide at quantiles 0.2, 0.4, and 0.8, all of which are positive. The absolute value of the coefficient is the largest at quantile 0.2, while the absolute value of the coefficient is the largest at quantile 0.8 among all significant results. The results indicate that OGD exerted a larger negative impact on the urban–rural income divide in regions with wider such divides before opening and a smaller negative impact on the divide in the regions where it was initially narrower. This provides more empirical evidence of the mechanisms through which OGD influences the divide, and also reduces concerns about reverse causality.

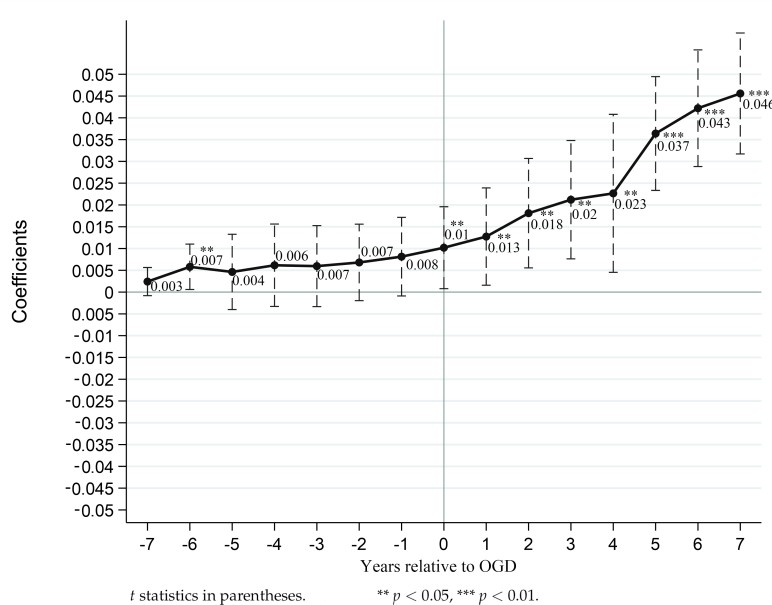

**Figure 3.** The dynamic impact of OGD on the Theil index of income inequality. The figure plots the impact of OGD on the Theil index of income inequality between urban and rural sectors according to the estimated coefficients and the corresponding standard error information from the above parallel trend test regression. We consider a 14-year window, spanning from 7 years before OGD until 7 years after. The dashed vertical lines represent 95% confidence intervals. If the vertical lines have an intersection with the X-axis, it indicates no significant difference (heterogeneous time trend) between the treatment group and the control group, and vice versa.

## 6. Conclusions

Beyond the enthusiasm for the potential advantages of the data revolution, there is growing concern regarding its possible detrimental effects. The unequal distribution, accessibility, and utilization of data exacerbate developmental disparities between individuals and groups. This complexity necessitates the advancement of theoretical and empirical frameworks. Building on the nonrival nature of data, we propose a model in which data serve as both an input and an output in the production process. Products created using data generate new data through consumption and usage, which are then fed back into production (e.g., self-driving cars; see Section 3.4). By constructing a theoretical model of this economic process and examining the role of open government data within it, we demonstrate the critical function of open government data and its inverted U-shaped impact on labor income inequality. Our empirical investigation provides evidence that the opening of government data in recent years has positively impacted the widening of the urban–rural income divide in China at the current stage. This finding is robust to an array of sensitivity analyses, and we find no evidence that reverse causality drives the results. Moreover, the impact of OGD on income distribution varies across cities with distinct initial economic characteristics. Due to the recent opening of government data in China over the past few years, the exact amount of data opened by each city has not been scientifically measured. Although we were unable to obtain precise data aligned with the continuous variables used in our theoretical models (as described in Section 3.4), we treated the occurrence of OGD as a binary variable in a DID regression for our empirical study. Despite this limitation, the empirical study is still meaningful as it provides initial insights into the potential impact of OGD policy on the urban–rural income divide at the early stages of its implementation in each city.

These findings align with the research conducted by Farboodi [43] and Jones and Tonetti [59]. Farboodi suggests that goods or services that heavily rely on data generate higher income compared to others due to the use of data for prediction and the subsequent

reduction in uncertainty, which enhances firm profitability. Similarly, Jones and Tonetti argue that data have the potential to generate substantial income, with data-intensive goods or services yielding higher income than other goods or services.

In our study, the causal relationship between OGD and the urban–rural income divide can be primarily explained by the disparity in the incorporation of data in the production and consumption of goods between the urban and rural sectors. The nonrival nature of data implies that those who generate and utilize data stand to gain significant benefits. Theoretically, a greater quantity of data results in higher returns. Regarding urban–rural income distribution, differences in the marginal production elasticity of capital between urban and rural areas, as well as variations in infrastructure, professional expertise, and digital firms, lead to rural areas generating and utilizing less data compared to urban areas during the early stages of economic development. This imbalance in economic competitiveness persists when government data become available, resulting in rural residents earning lower profits from data.

Furthermore, the causal relationship can be also explained by the disparity in the marginal production elasticity of urban and rural capital in China. Despite the government's efforts to eliminate it, the dual urban–rural system still exists. Currently, urban capital exhibits higher marginal production elasticity than rural capital. As demonstrated in our OGD and income inequality model in Section 3.4, this pre-existing disparity serves as one of the factors contributing to the positive relationship between OGD and the urban–rural income divide observed in our empirical study. From a data-centric perspective, we propose increasing the availability of data in relatively low-income areas. This could be achieved through the provision of comprehensive broadband infrastructure, transportation, and logistics in rural communities, enabling rural residents to access the digital world in their daily lives and work. Additionally, we recommend encouraging the development of information technology firms focused on the rural market and promoting digital agriculture. We believe that providing rural residents with opportunities to derive equal value from the rapid growth of data will gradually reduce the income gap with urban areas and promote equitable development in the digital age.

**Author Contributions:** Conceptualization, L.T. and J.P.; methodology, writing—original draft preparation, L.T.; writing—review and editing, J.P.; supervision, project administration, funding acquisition, J.P. All authors have read and agreed to the published version of the manuscript.

**Funding:** This research was funded by the Beijing Jiaotong University corporate campus cooperation program—Research on Intelligent Scheduling System of railway locomotive Wheel-set repair Line and its job creation and substitution effect, program No. B23SK00200.

**Institutional Review Board Statement:** Not applicable.

**Informed Consent Statement:** Not applicable.

**Data Availability Statement:** Online space for data storage is being applied for. All the data in the research will be shared.

**Conflicts of Interest:** The authors declare no conflict of interest. The funders had no role in the design of the study.

## Abbreviations

The following abbreviations are used in this manuscript:

| | |
|---|---|
| OGD | open government data |
| GAP | rural–urban income gap |

## Appendix A

In this appendix, we provide additional details and data supplemental to the main text.

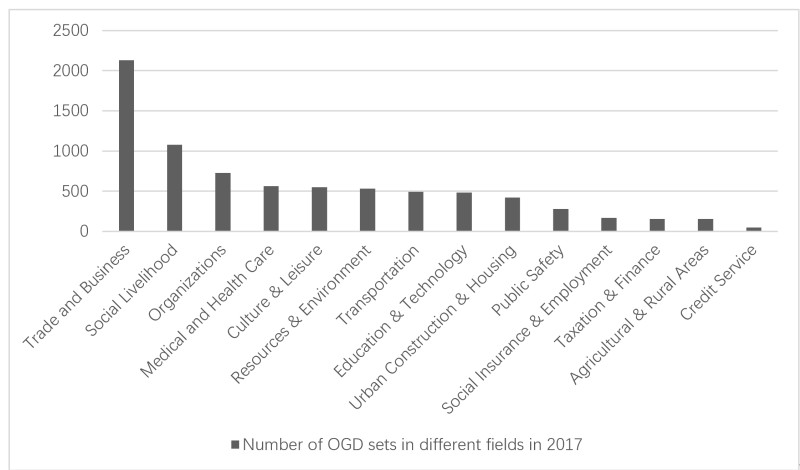

**Figure A1.** Number of OGD sets in different fields in 2017.

**Table A1.** Timing of OGD.

| City | Id | Year of Opening | City | Id | Year of Opening |
|---|---|---|---|---|---|
| Beijing | 1 | 2012 | Hena | 159 | 2018 |
| Tianjin | 2 | 2019 | Wuhan | 178 | 2015 |
| Wuhai | 30 | 2018 | Jingmen | 184 | 2016 |
| Haerbin | 62 | 2016 | Huanggang | 187 | 2019 |
| Jiamusi | 69 | 2019 | Changsha | 191 | 2016 |
| Shanghai | 74 | 2012 | Changde | 197 | 2019 |
| Jiangsu | 75 | 2019 | Yongzhou | 201 | 2019 |
| Najing | 76 | 2018 | Andong | 204 | 2016 |
| Moxi | 77 | 2014 | Anzhou | 205 | 2016 |
| Xuzhou | 78 | 2019 | Shaoguan | 206 | 2019 |
| Changzhou | 79 | 2019 | Shenzhen | 207 | 2016 |
| Suzhou | 80 | 2018 | Zhuhai | 208 | 2018 |
| Natong | 81 | 2019 | Shantou | 209 | 2019 |
| Lianyungang | 82 | 2019 | Foshan | 210 | 2014 |
| Huaian | 83 | 2019 | Jiangmen | 211 | 2018 |
| Yangzhou | 85 | 2018 | Zhanjiang | 212 | 2015 |
| Taizhou | 87 | 2019 | Maoming | 213 | 2019 |
| Suqian | 88 | 2019 | Zhaoqing | 214 | 2015 |
| Zhejiang | 89 | 2014 | Huizhou | 215 | 2018 |
| Ningbo | 91 | 2018 | Meizhou | 216 | 2016 |
| Huzhou | 94 | 2019 | Shanwei | 217 | 2019 |
| Bangbu | 104 | 2019 | Heyuan | 218 | 2019 |
| Maanshan | 106 | 2018 | Yangjiang | 219 | 2016 |
| Huangshan | 110 | 2019 | Qingyuan | 220 | 2019 |
| Fuyang | 112 | 2019 | Dongguan | 221 | 2016 |
| Liuan | 115 | 2018 | Zhongshan | 222 | 2018 |
| Xuancheng | 118 | 2018 | Chaozhou | 223 | 2019 |
| Fujian | 119 | 2019 | Jieyang | 224 | 2019 |
| Fuzhou | 120 | 2019 | Yunfu | 225 | 2019 |
| Shamen | 121 | 2019 | Naning | 227 | 2019 |
| Jiangxi | 129 | 2018 | Haina | 241 | 2019 |
| Fuzhou | 139 | 2019 | Sanya | 243 | 2019 |
| Shandong | 141 | 2018 | Sichuan | 245 | 2019 |
| Jina | 142 | 2018 | Chengdou | 246 | 2018 |
| Qingdao | 143 | 2015 | Luzhou | 249 | 2019 |
| Zibo | 144 | 2018 | Mianyang | 251 | 2019 |
| Zaozhuang | 145 | 2018 | Anyuan | 252 | 2019 |
| Dongying | 146 | 2018 | Suining | 253 | 2019 |

**Table A1.** *Cont.*

| City | Id | Year of Opening | City | Id | Year of Opening |
|------|-----|----------------|------|-----|----------------|
| Yantai | 147 | 2018 | Neijiang | 254 | 2019 |
| Weifang | 148 | 2018 | Yaan | 261 | 2019 |
| Jining | 149 | 2018 | Guizhou | 264 | 2016 |
| Taian | 150 | 2018 | Guiyang | 265 | 2017 |
| Weihai | 151 | 2018 | Liupanshui | 266 | 2019 |
| Rizhao | 152 | 2018 | Zunyi | 267 | 2019 |
| Laiwu | 153 | 2018 | Shanxi | 280 | 2018 |
| Linyi | 154 | 2018 | Ningxia | 306 | 2018 |
| Dezhou | 155 | 2018 | Yinchuan | 307 | 2018 |
| Liaocheng | 156 | 2018 | Danzuishan | 308 | 2018 |
| Binzhou | 157 | 2018 | Zhongwei | 311 | 2019 |
| Heze | 158 | 2018 | Tongren | 316 | 2018 |
| | | | Xinjiang | 312 | 2019 |

**Table A2.** Descriptive statistics.

| | N | Mean | Min | Max | Standard Deviation Across Cities | Within Cities | Within City Years |
|------|-----|------|-----|-----|-----|-----|-----|
| Theil index | 2121 | 0.083 | −0.357 | 0.283 | 0.023 | 0.045 | 0.017 |

**Table A3.** Quantile regression of urban–rural income Theil results.

| | Quantile 0.2 | Quantile 0.4 | Quantile 0.6 | Quantile 0.8 |
|------|-------------|-------------|-------------|-------------|
| OGD | 0.008 *** | 0.0021 * | 0.001 | 0.002 ** |
| | (3.44) | (1.84) | (1.19) | (2.42) |
| N | 409 | 400 | 409 | 416 |
| R2 | 0.8194 | 0.8356 | 0.7694 | 0.9661 |

\* $p < 0.1$, \*\* $p < 0.05$, \*\*\* $p < 0.01$.

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
