# Peer review of "Open Government Data and the Urban–Rural Income Divide in China: An Exploration of Data Inequalities and Their Consequences"

_sustainability, doi:10.3390/su15139867_

Round 1
Reviewer 1 Report
The manuscript analyses are very interesting and actually problematic. The theoretical background and research methods are adequate. The manuscript is adequately referenced. The results can be served as a base for future research.
The manuscript is original and relevant in the field of research. Authors adds broader analysis compared with other published material. Conculusions are consistent with the evidence and argument presented. References are appropriate.
The manuscript can be published in present form.
Author Response
Thank you for taking the time to review my article. I appreciate your positive feedback and I’m glad that you found the article to be valuable. Thank you again for your time and effort.
Reviewer 2 Report
Interesting topic and approach.
One of the basic conclusions is that `Our empirical investigation provides evidence that the opening of government data in recent years has positively impacted the widening of the urban-rural income divide in China at the current stage`. I think that the empirical analysis is not enough clear to prove that OGD influence the income gap between urban and rural areas. My suggestion is to improve the analysis in the Table 1: useful to give the VIF for each of the predictors in the table 1. The very high values of R2 (0.92) could be related to collinearity effects. If there are such effects, VIF could help to reveal what are the collinear predictors. If there are such effects, elimination of one or two collinear predictors could reduce the instability of the model. Ridge regression could help.
Author Response
Thank you for taking the time to review my article. Your comments and suggestions are very helpful and I appreciate your attention to detail. Please see the attachment for the revisions.

Reviewer 3 Report
The article addresses an important problem of relation between open government data availability and regional economic inequalities in China. The topic is interesting also for the policy implications.
The article is well structured. The content is presented in a clear and logical sequence.
Strengths of the article:
• Current research topic,
• A well-formulated research problem,
• Advanced analytical tools,
• Large population included in the study.
Remarks:
It seems that the article does not properly distinguish the method used in the analytical part. A methodological part should be added, in which econometric methods should be indicated along with source literature. Please provide brief information on the estimators used and the methods for assessing the quality of panel data models.
Please add the software used for calculations.
Author Response
Thank you for taking the time to review my article. I appreciate your feedback and I’m glad that you found the article to be valuable. Your comments and suggestions are very helpful, and my revisions are as follows.
Point 1: It seems that the article does not properly distinguish the method used in the analytical part. A methodological part should be added, in which econometric methods should be indicated along with source literature. Please provide brief information on the estimators used and the methods for assessing the quality of panel data models.
Please add the software used for calculations.
Response 1: The following part has been added to the Methodology part "Since government data were opened in different years in different cities, it provides a natural setting for an event study to use multiperiod difference-in-differences (DID) specification to assess the relation between OGD and urban--rural income divide \cite{callaway2020difference}. As an extension of the standard DID method, multiperiod DID allows for more than two periods of data. It is used to estimate the causal effect of a policy intervention that begins at different times in different cities. In our scenario, the DID method compares the change in outcomes over time between the treatment group (cities that opened government data) and the control group (cities that did not open government data). (All the calculations were run in Stata.)"
Reviewer 4 Report
In my opinion, the following claims can be made against the article:
1. It is necessary to subtract the text. There are a lot of errors
2. The formulas given in the article are based on differential and integral calculus. It is not entirely clear how these formulas were applied to calculations on discrete statistical data.
3. The graphs shown in Figures 1-3 are not indicative. In Figures 1-2, you need to add trends to explain the relationships. Figure 3 needs to be supplemented with actual data. Also, explanatory information should be added to the figures so that the patterns assumed by the author can be understood.
4. The main conclusion of the article is not fully understood, how the data increase the gap between rural and urban areas. It is quite possible that the situation is reversed: the level of development of the territory determines the amount of data collected. Either these two trends are the result of some third factor. A deeper qualitative analysis is needed to explain this causal relation.
-
Author Response
Dear Reviewer,
Thank you for taking the time to review my article. Your suggestions have helped me to clarify my arguments and strengthen my conclusions.
I have addressed each of your comments and suggestions in the attachment. Specifically, I have inserted all the changes into the manuscript. I hope that these changes address your concerns.
